# Associations of Protein Molecular Structures with Their Nutrient Supply and Biodegradation Characteristics in Different Byproducts of Seed-Used Pumpkin

**DOI:** 10.3390/ani12080956

**Published:** 2022-04-07

**Authors:** Yang Li, Qinghua Wu, Jingyi Lv, Xiaoman Jia, Jianxu Gao, Yonggen Zhang, Liang Wang

**Affiliations:** 1College of Animal Science and Technology, Northeast Agricultural University, Harbin 150030, China; liyang1405053@sina.com (Y.L.); wqh150339@sina.com (Q.W.); 18846440487@163.com (J.L.); xman1008@163.com (X.J.); 18846176952m@sina.cn (J.G.); 2Research Institute of Applied Technologies, Honghe University, Mengzi 661199, China

**Keywords:** byproducts of seed-used pumpkin, protein molecular structures, nutritive values, in situ rumen degradation kinetics, relationship

## Abstract

**Simple Summary:**

At present, the world is faced with a shortage and rising prices of high-quality protein feed. Adequate utilization of unconventional feeds can alleviate this problem. China is the largest producer and exporter of seed-used pumpkin, and many byproducts of seed-used pumpkin need to be effectively utilized. In this study, we analyzed nutrient supply and biodegradation characteristics, and correlation and regression equations between them and protein molecular structure were established. This research provides a data basis for the utilization of byproducts of seed-used pumpkin and establishes a new method for evaluating the nutritional value of unconventional feeds.

**Abstract:**

The purpose of this experiment was to explore the relationship of protein functional groups (including amide I, amide II, α-helix, and β-sheet) in byproducts of seed-used pumpkin (pumpkin seed cake, pumpkin seed coat, and seed-used pumpkin flesh) with their nutrient profiles and biodegradation characteristics. The experiment was designed to use conventional chemical analysis, combining the Cornell Net Carbohydrate and Protein System (CNCPS) and nylon bag technology to assess the nutritional value and biodegradation characteristics of seed-used pumpkin byproducts. Fourier transform infrared spectroscopy (FTIR) was used to analyze the protein molecular structure properties of byproducts of seed-used pumpkin. In this study, we also applied correlation and regression analysis. The results showed that different byproducts of seed-used pumpkin had different in situ biodegradation, nutrient supplies, and spectral structures in the protein region. Among the byproducts of seed-used pumpkin, acid detergent-insoluble crude protein (ADICP) and neutral detergent-insoluble crude protein (NDICP) contents of the pumpkin seed coat were the highest, resulting in the lowest effective degradabilities (EDs) of dry matter and crude protein. The crude protein (CP) ED values were ranked as follows: pumpkin seed cake > seed-used pumpkin flesh > pumpkin seed coat. Significant differences were observed in the peak areas of amide I and amide II and the corresponding peak heights in the two peak areas in the molecular structure of the protein. The peak areas of amide I and amide II and the corresponding peak heights were at the highest levels for pumpkin seed cake, whereas there was no significant difference between the pumpkin seed coat and seed-used pumpkin flesh. Similarly, the peak heights of α-helices and β-sheets were highest for pumpkin seed cake. Correlation and regression results indicated that amide I and amide II area and height, α-helix and β-sheet heights, and area ratios of amide I: amide II, as well as the height ratios of amide I: amide II, and α-helices: β-sheets effectively estimated nutrient supply and that the height ratio of α-helices: β-sheets was mostly sensitive to biodegradation characteristics in different byproducts of seed-used pumpkin. There were significant differences in CP chemical composition and digestibility of different byproducts of seed-used pumpkin that were strongly related to the changes in protein molecular structures.

## 1. Introduction

Today, many countries are facing feed shortages, and high feed costs affect the economic benefits of pastures. The use of agricultural byproducts is an effective way to solve this problem. Seed-used pumpkin, a general term for pumpkins of which the seeds are the main edible organs or processing objects, belongs to the Cucurbitaceae family. China is one of the most important countries producing and exporting pumpkin seeds globally. The planting area and total output of pumpkin seeds in China rank first in the world [1]; therefore, China produces a large number of byproducts of seed-used pumpkin (seed-used pumpkin flesh, pumpkin seed coat, and pumpkin seed cake) every year. Using such byproducts for animal feeding is a means of recycling; otherwise, it would accumulate or be directly incinerated, causing severe environmental pollution and waste resources.

As reported previously, the protein content of pumpkin seed cake is higher than 500 g/kg (dry matter, DM), which is higher than that of soybean meal [2,3,4], and can provide a sufficient nitrogen source for ruminants. In addition, pumpkin seed cake is rich in limiting amino acids (lysine and methionine), and there have been studies on feeding pumpkin seed cake to ruminants [4]. The pumpkin seed coat is the outer skin of pumpkin seeds obtained by soaking, peeling, and drying the pumpkin seeds. It is considered a short fiber roughage rich in crude protein (162.6 g/kg DM). At present, no research has been reported on the use of pumpkin seed coats as animal feed. Seed-used pumpkin flesh is rich in nutrients, such as protein (114.0 g/kg DM), starch (51.6–102.4 g/kg DM), soluble sugar (224.6–474.3 g/kg DM), crude fiber (130.5–185.7 g/kg DM) [1], and bioactive molecules (carotenoids, 0.20–19.57 mg/100 g, wet basis) [5], which can provide sufficient energy and antioxidants for ruminants [6,7]. The byproducts of seed-used pumpkin are rich in protein, and their respective nutritional characteristics are different, which are valuable feed resources for ruminants. Research on the nutritional composition and biobehavioral features of byproducts of seed-used pumpkin is essential for the rational use of byproduct feeds. However, few reports can be found on either the wet chemical analysis or spectral structural investigation of the byproducts of seed-used pumpkin. Research has revealed that the correlation between the molecular structure characteristics of byproducts of seed-used pumpkin and nutritional status would help to improve the feed utilization efficiency of byproducts of seed-used pumpkin in ruminant production [8,9]. It is of great significance to alleviate the shortage of feed resources and improve the economic benefits of pastures.

Compared with traditional wet chemistry analysis, Fourier transform infrared spectroscopy (FTIR) has been proven to be a bioanalytical technique that can quickly, noninvasively, and directly reveal the molecular structure of feeds [8]. With the widespread application of the FTIR technique, the technology is not limited to the study of molecular structural characteristics of cereal grains [10,11], oilseed [12], and forage [13,14]; spectroscopic studies of agricultural processing byproducts [15] have also been conducted. The protein spectral region comprises amide I, amide II, α-helix, and β-sheet characteristic regions. Changes in the molecular structure information of the feed spectrum obtained by FTIR analysis affect feed utilization and nutritional value [9,11]. Therefore, it is necessary to use FTIR to analyze the byproducts of seed-used pumpkin. Conventional chemical analysis methods, combined with the Cornell Net Carbohydrate and Protein System (CNCPS) and nylon bag technology, were used in this study, aimed to comprehensively investigate the chemical properties, nutrient supply, and biodegradation properties of byproducts of seed-used pumpkin. Using FTIR to analyze the inherent spectral structure of byproduct protein functional groups, correlation and regression analyses were carried out to determine the relationship between nutritional value and spectral parameters and enrich the mid-infrared spectral database while developing and evaluating new methods for unconventional feed.

## 2. Materials and Methods

### 2.1. Sample Collection

Seed-used pumpkin collected from Heilongjiang Hexing Agricultural Products Company, Ltd. in Shuangyashan city, China, in early October 2018 was used in this study. In this study, three varieties (Yinhui 1, Yinhui 2, and Jinhui 1) were selected as replicates. Fifty seed-used pumpkins were selected for each variety, and each seed-used pumpkin was manually separated into three parts (pumpkin seed, pumpkin seed coat, and seed-used pumpkin flesh). Additionally, pumpkin seed cake was obtained as a byproduct from oil production through pumpkin seed crushing by a physical cold-pressing method. Pumpkin seeds were processed using a spiral oil extractor (KYL-380 oil extractor) with a controlled temperature of 70 °C, moisture content of 7.5%, rotation rate of 36 r/min, and oil extraction rate of about 47%. Pumpkin seed cake was obtained after oil extraction of the pumpkin seeds. The samples (*n* = 3) were collected from corresponding byproducts, which were used as replicates. After drying in a forced-air oven at 55 °C for 48 h, all samples were chemically analyzed by a 1.0 mm screen and spectrally scanned twice by a 0.25 mm screen.

### 2.2. Chemical Analysis

The AOAC method [16] was used to determine the crude protein (CP, AOAC 984.13) and dry matter (DM, AOAC 930.15) contents of the byproduct samples. The contents of neutral detergent fiber (NDF) and acid detergent fiber (ADF) were measured according to the filtration method [17] using an Ankom 220 fiber analyzer (Ankom Technology Corp., Macedon, NY, USA). The contents of neutral detergent-insoluble CP (NDICP) and acid detergent-insoluble CP (ADICP) in NDF and ADF filter residue were analyzed by the Kjeldahl-N method [17]. Non-protein nitrogen (NPN) and total soluble CP (SCP) were analyzed using a previously published method [18]. The Cornell Net Carbohydrate and Protein System (CNCPS) [19] partitions CP into five subfractions, namely PA (nonprotein nitrogen (NPN)), PB_1_ (rapidly degradable protein), PB_2_ (moderately degradable protein), PB_3_ (slowly degradable protein), and PC (undegradable protein).

### 2.3. In Situ Rumen Degradation

The in situ rumen degradation characteristics of DM and CP of the three byproducts of seed-used pumpkin were analyzed according to a method described previously [20]. Three Holstein cows ruminally fistulated and housed in individual stalls (BW = 625 ± 15.2 kg, mean ± SD) were used for the in situ incubations. The Northeast Agricultural University Animal Science and Technology College Animal Care and Use Committee (Protocol number: NEAUEC20181007) approved all experimental procedures involving animals. The cows were fed 2 times daily (8:00, 16:00) with a total mixed ration (TMR) for a total DMI of 15 g/kg of BW and provided free access to fresh water for drinking. The TMR (g/kg of DM) consisted of Chinese wildrye (428.7 g/kg), corn silage (158 g/kg), and concentrate (413.3 g/kg) according to the NRC requirements. In this study, we weighed 7 g of the three byproduct samples of seed-used pumpkins and put each into a numbered nylon bag (10 × 20 cm, pore size of 50 μm; Ankom Technology, Macedon, NY, USA), which was tied with nylon rope. All of the nylon bags were tied to the end of a 40 cm polyester mesh bag and then placed in the ventral sac of the rumen through the ruminal fistula to incubate for 0, 2, 4, 8, 12, 16, 24, 36, and 48 h according to the “all in/gradual out” schedule. The samples were incubated in two replicate runs. After incubation, all bags were rinsed thoroughly with cold tap water until the washed water was clear. The bags were then dried at 55 °C for 48 h and weighed, and then the residues were ground and passed through a 1 mm screen for DM and CP analysis. The in situ degradation constants were estimated using a nonlinear model described previously [21] as *p* = *a* + *b* (1 − exp^(−ct)^), where *p* = the rate of disappearance at time t (h): 

*a* = the rapidly degradable fraction (the soluble fraction); 

*b* = the slowly degradable fraction; 

*c* = rate at which degradation occurred (*c* > 0);

*a* + *b* = potentially degradable fraction in rumen degradation. 

The effective degradability (*ED*) was calculated, assuming a passage rate (*kp*) of 46 g/kg h^−1^ (pumpkin seed coat) or 60 g/kg h^−1^ (pumpkin seed cake and seed-used pumpkin flesh), by the equation:*ED* = *a* + *bc*/(*c* + *kp*), 
where *a*, *b*, and *c* are the constants described above.

### 2.4. Spectral Data Collection and Analysis

The protein spectral data of seed-used pumpkin byproducts (including pumpkin seed coat, pumpkin seed cake, and seed-used pumpkin flesh) were collected by FTIR spectroscopy (Bruker ALPHA-T, Berlin, Germany) in the Laboratory of Chemical Molecular Structure Analysis, College of Science, Northeast Agricultural University (Harbin). The mid-IR spectra were scanned 128 times in the range of approximately 4000~400 cm^−1^ with a spectral resolution of 4 cm^−1^. In addition, each sample was scanned five times.

OMNIC 8.2 software (Thermo Nicolet Co., Madison, WI, USA) was used to analyze the spectral data of all samples. The baseline position of the protein was determined in the software, as well as the amide Ⅰ band area (baseline: ca.1712~1576 cm^−1^) and peak height position (ca. 1638 cm^−1^) and the amide Ⅱ band area (baseline: ca.1536~1488 cm^−1^) and peak height position (ca. 1528 cm^−1^). Fourier self-deconvolution (FSD) [22] or the second derivative function (2nd derivative) [23] was used to determine the position of the α-helix peak height (ca. 1650 cm^−1^) and β-fold peak height (ca. 1617 cm^−1^) in amide Ⅰ. Then, the peak height and area were recorded and statistically analyzed. The typical ATR-FT/IR molecular spectra of different byproducts of seed-used pumpkin are shown in Figure 1.

### 2.5. Statistical Analyses

The protein nutritional values, rumen degradation parameters, and spectral values of byproducts of seed-used pumpkin (pumpkin seed cake, seed pumpkin coat, and seed-used pumpkin flesh) were analyzed using the ANOVA procedure of the SAS software system (version 9.4, SAS Institute Inc., Cary, NC, USA). The statistical model was as follows: 

*Y_ij_* = *μ* + *T_i_* + *E_i_*;

*Y_ij_* = dependent variable;

*μ* = overall mean;

*T_i_* = treatment effect;

*E_i_* = error term.

Using the Pearson correlation method and the PROC CORR program of SAS 9.4, we analyzed the correlation between the protein nutritional values and protein-related molecular-structure spectral parameters in different coproducts of seed-used pumpkin. Finally, regression analysis was performed using the PROC REG program of SAS 9.4.

## 3. Results

### 3.1. Chemical Composition and CNCPS Protein Components of Seed-Used Pumpkin Byproducts

Table 1 shows that the contents of CP, SCP, NPN, ADICP, and NDICP of seed-used pumpkin byproducts were significantly different (*p* < 0.05). Among these, pumpkin seed cake had the highest CP and SCP contents (553.1 g/kg DM and 119.8 g/kg DM, respectively). The highest ADICP content was found in the pumpkin seed coat, which was significantly higher than that of the pumpkin seed cake and seed-used pumpkin flesh (*p* < 0.05), indicating that the pumpkin seed coat was not easily digested in the rumen. Seed-used pumpkin flesh had the highest NPN content, which was significantly higher than that of pumpkin seed cake and pumpkin seed coat (*p* < 0.05). From the perspective of CNCPS protein components, there were significant differences in the contents of PA, PB_1_, PB_2_, and PC in pumpkin seed cake, pumpkin seed coat, and seed-used pumpkin flesh (*p* < 0.05). Among these, seed-used pumpkin flesh had the highest PA content, which was significantly higher than that of pumpkin seed cake and pumpkin seed coat (*p* < 0.05), indicating that seed-used pumpkin flesh can be rapidly degraded in the rumen. Pumpkin seed cake had the lowest PC content (3.2 g/kg CP), indicating that pumpkin seed cake can provide more nitrogen sources for ruminants, whereas pumpkin seed coat had the highest content (498.4 g/kg CP) because its ADICP content was higher, which could make it difficult to digest in the rumen.

### 3.2. Rumen Degradation Parameters of DM and CP of Seed-Used Pumpkin Byproducts

Table 2 shows that the kinetic parameters of rumen degradation of DM and CP were different (*p* < 0.05) for different seed-used pumpkin byproducts (pumpkin seed cake, pumpkin seed coat, and seed-used pumpkin flesh). Among these, *a*, *b*, *c*, *a* + *b*, and *ED* of DM and CP in pumpkin seed coat were significantly lower than those of pumpkin seed cake and seed-used pumpkin flesh (*p* < 0.05). Compared with the pumpkin seed cake, seed-used pumpkin flesh had more rapidly degradable DM and CP (*p* < 0.05). However, the *b* and *c* values were higher in pumpkin seed cake than in seed-used pumpkin flesh (*p* < 0.05), so it is certain that they had a significant difference in their effective degradability of DM and CP. In the current research, the *b*, *c,* and *ED* values were ranked as pumpkin seed cake > seed-used pumpkin flesh > pumpkin seed coat; the *a* value was ranked as seed-used pumpkin flesh > pumpkin seed cake > pumpkin seed coat; and the *a* + *b* value of pumpkin seed coat was lower in the two other byproducts.

### 3.3. Spectral Parameters of the Protein Molecular Structure of Seed-Used Pumpkin Byproducts

As shown in Table 3 and Figure 1, there were significant differences in protein molecular structure and spectral parameters from pumpkin seed cake, pumpkin seed coat, and seed-used pumpkin flesh. The peak height ratios of the α-helix and β-sheet of the pumpkin seed coat were significantly lower than those of the pumpkin seed cake and seed-used pumpkin flesh (*p* < 0.05). Additionally, the peak height ratio of the amide Ⅰ band and amide Ⅱ band of seed-used pumpkin flesh was significantly higher than that of pumpkin seed cake and pumpkin seed coat (*p* < 0.05). In addition, the peak height and peak area of the amide Ⅰ band, the peak height and peak area of the amide Ⅱ band, and the peak height and ratio of α-helices and β-sheets of pumpkin seed cake were significantly higher than those of pumpkin seed coat and seed-used pumpkin flesh (*p* < 0.05), indicating that the nutritional value of pumpkin seed cake protein is high.

### 3.4. Correlations between the Protein Molecular Structure of Seed-Used Pumpkin Byproducts and Their Conventional Chemical Components, CNCPS Protein Components, and Rumen Degradation Characteristics

In Table 4, the results of the correlations between the spectral parameters of seed-used pumpkin byproducts and their nutritional value, CNCPS protein subfractions, and in situ rumen degradation characteristics are summarized. The CP, SCP, PB_1_, and PB_2_ had a strong positive correlation (*r* = 0.73~0.97; *p* < 0.05) with amide I and II height and area. However, a negative correlation (r = −0.69~−0.82; *p* < 0.05) was also found between DM, CP, and NDICP; the area ratio of amide I and amide II and the content of CP and PB_2_ correlated with the peak height ratios of amide I and amide II (*r* = −0.74~−0.83; *p* < 0.05). The peak height ratios of α-helices and β-sheets were extremely significantly negatively correlated with DM, ADICP, and PC (*r* = −0.95~0.98, *p* < 0.01) and extremely significantly positively correlated with SCP and PB_2_ (*r* = 0.83~0.94, *p* < 0.01). Among them, the *c* values of DM (*r* = 0.85~0.93; *p* < 0.05) and CP (*r* = 0.81~0.92; *p* < 0.05) rumen degradation kinetics had a strong positive correlation with peak height, as well as the areas of amides Ⅰ and Ⅱ, the peak heights of α-helices and β-sheets, and the sum of the peak areas of amides Ⅰ and Ⅱ. The peak height ratio of α-helices and β-sheets had a very significant positive correlation with *a*, *b*, *a* + *b*, *c*, and their effective DM and CP degradation rates (*r* = 0.83~0.97, *p* < 0.01).

### 3.5. Regression Relationship between the Protein Molecular Structure of Seed-Used Pumpkin Byproducts and Their Nutritional Value and Rumen Degradation Characteristics

Table 5 shows multiple regression analyses used to select spectral parameters related to proteins to predict the protein subfraction, nutritional value, and rumen degradation characteristics of the seed-used pumpkin byproducts. It can be seen from the results that the protein chemical profiles, protein subfractions using the CNCPS system, and in situ DM and CP rumen degradation kinetics were mainly associated with amide I and amide II area and height, α-helix and β-sheet height, area ratios of amide I: amide II, and height ratios of amide I: amide II and α-helix: β-sheet, with the height ratio of α-helix: β-sheet showing the greatest influence on the biodegradation characteristics of different byproducts of seed-used pumpkin.

The height ratio of α-helix: β-sheet showed a strong ability to forecast the contents of DM (R^2^ = 0.995), CP (R^2^ = 0.973), SCP (R^2^ = 0.993), ADICP (R^2^ = 0.8796), PA (R^2^ = 0.980), PB_2_ (R^2^ = 0.972), and PC (R^2^ = 0.996) by the regression equation results. The kinetics of in situ degradation of DM and CP (*p* < 0.001; R^2^ = 0.852–0.997) in the rumen were predicted from the spectral features within protein regions.

## 4. Discussion

The price fluctuation of feed and animal products is a characteristic of animal husbandry, and the cost of animal products directly determines the economic success of a farm [24]. The tight supply–demand relationship of high-quality roughage, the shortage of high-quality protein resources, and high feeding cost have always been bottlenecks in the efficient production of China’s animal husbandry industry. One of the major problems facing the cattle industry is the shortage of high-quality protein resources in China. Thus, the development of new protein feed resources is of great significance to alleviate the shortage of feed resources and promote the development of the dairy industry. Soybean meal is high in protein and contains essential amino acids for animals; therefore, it is a commonly used protein source in animal feed [25]. However, the price of soybean meal fluctuates and has been on the rise; therefore, to maximize profit, other protein resources are being used as an alternative protein feed sources for ruminants [26]. Pumpkin seed cake, a secondary product of pumpkin seed oil processing, often serves as a protein source for ruminant feed [4]. Many popularly use oilseed feed ingredients, including soybean meal, which has a low crude protein concentration (crude protein, CP, 598 g/kg) [2,3,27], high lysine (32 g/kg) and methionine (18 g/kg) contents, and improved palatability of concentrates for ruminants [28]. In this study, the protein content of pumpkin seed cake was as high as 553.1 g/kg, which was significantly higher than that of pumpkin seed coat and seed-used pumpkin flesh, similar to the results of previous studies [4]. The SCP and NPN contents of pumpkin seed cake and seed-used pumpkin flesh were significantly higher than those of pumpkin seed coat, indicating that the two feeds can be rapidly degraded in the rumen. Because ADICP and NDICP have difficulty providing usable amino acids for the rumen, the increase in the content of ADICP and NDICP causes the feed to be indigestible in the rumen [29]. This experiment’s higher PA content of seed-used pumpkin flesh was due to the higher NPN content. The decrease in PB_2_ content is related to the decrease in the effective degradation rate of CP in rumen [9]. In this experiment, the PB_2_ content of the pumpkin seed coat was significantly lower than that of the pumpkin seed cake and seed-used pumpkin flesh, so the rumen degradation rate must be lower than that of the two feeds. The high content of SCP in pumpkin seed cake is the reason why its PB_2_ content was significantly higher than that of the other two feeds. Pumpkin seed cake and seed-used pumpkin flesh have more protein available for rumen fermentation because they contain higher PA, PB_1_, and PB_2_ subfractions than pumpkin seed coat [30]. PC is not degradable in the rumen and contains lignin, tannin, and Maillard reaction products. At present, there are few studies on pumpkin seed coat, and its high content of insoluble protein makes its PC content significantly higher than that of the other two feeds. In research from other groups, pumpkin seed cake has been shown to completely replace soybean meal in a dairy goat diet without reducing milk production or drastically changing fatty acid composition, which may have commercial or human health benefits [4]. Therefore, it is necessary to perform an in-depth study on the feasibility of the development of byproducts of seed-used pumpkin as feed resources and the effect of its application in ruminant production.

The content of SCP and NPN in pumpkin seed cake and seed-used pumpkin flesh was significantly higher than that of the pumpkin seed coat, resulting in rapidly degradable fractions in rumen degradation of pumpkin seed cake and seed-used pumpkin flesh that were also higher than those of pumpkin seed coat [31]. Pumpkin seed cake and seed-used pumpkin flesh can provide more rapidly degradable protein for rumen microorganisms, and this also shows that the potentially degradable fraction in rumen degradation of both is significantly higher than that of pumpkin seed coat. Because the fiber content of pumpkin seed coat is higher than that of pumpkin seed cake and seed-used pumpkin flesh, the combination with protein in the production and processing treatment resulted in a significant increase in the content of NDICP and ADICP in the pumpkin seed coat. The high content of NDICP and ADICP causes the rumen effective degradation rate of the pumpkin seed coat to decrease. However, because the neutral detergent-insoluble protein can be digested and absorbed by the small intestine after passing through the rumen, pumpkin seed coat and pumpkin seed cakes with a high content of neutral detergent-insoluble protein can also provide rumen bypass protein for dairy cows [32]. Because the SCP of pumpkin seed cake and seed-used pumpkin flesh is significantly higher than that of pumpkin seed coat, its NDICP and ADICP are lower than those of the pumpkin seed coat, resulting in the effective protein degradation rate of pumpkin seed cake and seed-used pumpkin flesh being higher than that of pumpkin seed coat. Therefore, pumpkin seed cake and seed-used pumpkin flesh can provide high-yield lactating cows with more easily degradable protein during production.

The protein infrared spectrum mainly includes amide I and amide II. The spectral region of amide I is at 1600–1690 cm^−1^, and the spectral region of amide II is mainly at 1480–1575 cm^−1^ [33]. Previous studies have proven that changes in protein molecular structure cause protein solubility and the absorption and utilization of feed protein by rumen microorganisms and proteolytic enzymes [34]. The difference in the number of protein functional groups is shown by the peak height and peak area of the protein amide I band and the amide II band, and their ratio indicates the change in protein structure composition [35]. In this experiment, the peak height and peak area of amide I and the peak height and peak area of amide II in pumpkin seed cake were significantly higher than those of pumpkin seed coat and seed-used pumpkin flesh, indicating that the nutritional value of protein in pumpkin seed cake is higher. The secondary structure of proteins primarily includes α-helices, β-sheets, and small amounts of β-turns and random coils [32]. Even if the protein content in the feed is similar, if the ratio of α-helices to β-sheets in the protein secondary structure is different, its nutritional value and protein availability may also be different [32,36]. In this experiment, the peak heights and ratios of α-helices and β-sheets of pumpkin seed cake were significantly higher than those of pumpkin seed coat and seed-used pumpkin flesh, indicating the nutritional value and availability of protein in pumpkin seed cake is higher than that of pumpkin seed coat and seed-used pumpkin flesh. Therefore, studying the differences in the internal molecular structure of feed is of great significance for quickly understanding its chemical composition, digestion, and utilization.

Studies have proven a correlation between the molecular structure of feed protein and its nutritional value, and protein spectral parameters can be used to predict the nutritional value and digestibility of feed [9,37]. Xin et al. (2020) found that crude protein is significantly positively correlated with the area and height of amide Ⅰ and amide Ⅱ and with α-helix and β-sheet height and ratio [38]. Previous studies also found that the protein secondary structure height of amide Ⅰ and amide Ⅱ and the height of α-helices and β-sheets have a significant positive correlation with SCP [39]. In this experiment, the height of amide Ⅰ and amide Ⅱ and the height of α-helices and β-sheets were also significantly positively correlated with PB_1_ in CNCPS protein components. Li et al., (2016) reached the same conclusion as this experiment [23]. The spectral structure of the feed protein molecule is closely related not only to the nutrient composition of the protein but also to the degradation characteristics of the feed in the rumen. Xin et al., (2020) found that in corn kernel and corn coproducts, there are significant negative correlations between the area and height of amides I and II, the area and height ratio of amides I and II, and the rate of degradation [38], which is consistent with the results of this experiment. The results of this experiment show a positive correlation between the ratio of H_α-helix and H_β-sheet in the secondary structure of the protein in seed-used pumpkin byproduct and EDCP, which is also consistent with the test results of Xin et al. (2020) [38]. From the above experimental results, it can be seen that the peak heights of α-helices and β-sheets of protein secondary structure and their ratios are good predictors of protein nutritional value and digestive behavior.

Changes in the molecular spectral structure of feed affects the nutritional value, digestive behavior, and utilization of the feed [11]. At present, there have been no reports on whether the molecular spectral information of byproducts of seed-used pumpkin can be employed to predict its biological behavior and nutritional value. Li et al. (2016) determined that the protein molecular structure of canola meal and soybean meal is related to their nutritional value and rumen degradation characteristics [23]. In addition, regression equations can be constructed through correlation, and the best regression equation has a coefficient of determination/R^2^ as high as 0.99. According to Yu et al. (2003), the ratio of spectral parameters can more accurately predict feed quality and nutritional value [40]. According to the correlation and regression equations, the peak area ratio of amide Ⅰ to amide Ⅱ and the peak height ratio of α-helices to β-folds can effectively estimate the protein nutrients and rumen degradation parameters of byproducts of seed-used pumpkin. Through the above research, we can conclude that it is feasible to use protein spectral parameters to evaluate the nutritional value of feed protein; there is a need to establish a large number of databases to more accurately predict the nutritional value of feed.

## 5. Conclusions

There were significant differences in the protein chemical profile, CNCPS subfraction, and rumen degradation kinetics of seed-used pumpkin byproducts. Among these byproducts, seed-used pumpkin flesh and pumpkin seed cake were shown to have high nutritional value and the ability to provide a nitrogen source for ruminants. The significant differences in the chemical profile of CP and the degradability of byproducts of seed-used pumpkin were closely related to the changes in the molecular spectral parameters of the protein. The regression equation shows that the height ratio of α-helices to β-sheets was a strong predictor of the protein chemical profile and degradability. It was initially proven that FTIR could be used to evaluate the nutritional supply and biodegradation characteristics of different byproducts of seed-used pumpkin.

## Figures and Tables

**Figure 1 animals-12-00956-f001:**
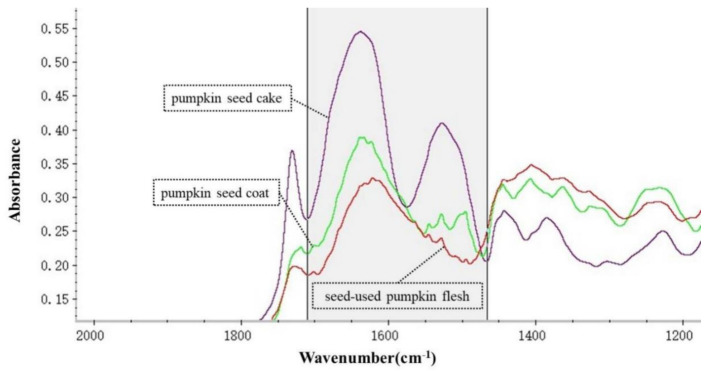
Typical molecular vibrational spectrum of different byproducts of seed-used pumpkin in the mid-IR within the protein amide region.

**Table 1 animals-12-00956-t001:** Chemical profiles and protein subfractions in different byproducts of seed-used pumpkin.

	Pumpkin Seed Cake	Pumpkin Seed Coat	Seed-Used Pumpkin Flesh	SEM ^1^	*p*
Chemical composition (g/100 g DM) ^2^					
DM	95.36 ^a^	87.84 ^c^	89.52 ^b^	0.016	<0.0001
CP	55.31 ^a^	16.26 ^b^	11.40 ^c^	0.386	<0.0001
SCP	11.98 ^a^	3.55 ^c^	6.51 ^b^	0.337	<0.0001
NPN	4.17 ^b^	2.25 ^c^	5.48 ^a^	0.377	0.0027
ADICP	0.18 ^c^	8.10 ^a^	0.87 ^b^	0.083	<0.0001
NDICP	6.82 ^b^	9.92 ^a^	2.64 ^c^	0.363	<0.0001
subfractions of protein (g/100 g CP, using the CNCPS) ^3^					
PA	7.53 ^c^	13.82 ^b^	47.92 ^a^	1.453	<0.0001
PB_1_	14.13 ^a^	8.01 ^b^	9.09 ^b^	1.312	0.0347
PB_2_	66.02 ^a^	17.15 ^b^	19.77 ^b^	0.958	<0.0001
PB_3_	12.01	11.18	15.59	1.276	0.1042
PC	0.32 ^c^	49.84 ^a^	7.64 ^b^	0.614	<0.0001

^1^ SEM, standard error of the mean. Means with the different letters in the same row are significantly different (*p* < 0.05). ^2^ DM, dry matter; CP, crude protein; SCP, soluble crude protein; NPN, non-protein nitrogen; ADICP, acid detergent-insoluble crude protein; NDICP, neutral detergent-insoluble crude protein; ^3^ PA, non-protein nitrogen; PB_1_, rapidly degraded protein; PB_2_, intermediately degraded protein; PB_3_, slowly degraded protein; PC, bound protein.

**Table 2 animals-12-00956-t002:** In situ rumen degradation kinetics in different byproducts of seed-used pumpkin.

	Pumpkin Seed Cake	Pumpkin Seed Coat	Seed-Used Pumpkin Flesh	SEM ^1^	*p*
in situ dry matter (DM) rumen degradation kinetics ^2^					
*a* (g/kg)	565 ^b^	235 ^c^	596 ^a^	5.8	<0.0001
*b* (g/kg)	391 ^a^	140 ^b^	385 ^a^	9.6	<0.0001
*a* + *b* (g/kg)	957 ^b^	375 ^c^	982 ^a^	6.0	<0.0001
*c* (g/kg h^−1^)	142 ^a^	88.5 ^c^	109 ^b^	3.77	0.0002
*ED* (g/kg)	840 ^a^	377 ^b^	844 ^a^	3.6	<0.0001
in situ dry matter (CP) rumen degradation kinetics					
*a* (g/kg)	510 ^b^	271 ^c^	566 ^a^	6.4	<0.0001
*b* (g/kg)	441 ^a^	122 ^c^	398 ^b^	10.5	<0.0001
*a* + *b* (g/kg)	951 ^a^	394 ^b^	964 ^a^	6.3	<0.0001
*c* (g/kg h^−1^)	174 ^a^	51.5 ^c^	108 ^b^	6.23	<0.0001
*ED* (g/kg)	837 ^a^	336 ^c^	822 ^b^	3.3	<0.0001

^1^ SEM, standard error of the mean. Means with the different letters in the same row are significantly different (*p* < 0.05). ^2^
*a*, rapidly degradable fraction in rumen degradation; *b*, slowly degradable fraction in rumen degradation; *a* + *b*, potentially degradable fraction in rumen degradation; *c*, degradation rate of the slowly degradable fraction; *ED*, effective degradability of the incubated samples.

**Table 3 animals-12-00956-t003:** Molecular structural characteristics of protein in different byproducts of seed-used pumpkin.

	Pumpkin Seed Cake	Pumpkin Seed Coat	Seed-Used Pumpkin Flesh	SEM ^1^	*p*
molecular spectral features ^2^					
A_Amide I	26.6 ^a^	11.0 ^b^	12.5 ^b^	1.54	0.0007
A_Amide II	14.0 ^a^	2.56 ^b^	0.70 ^b^	0.77	<0.0001
A_Amide I + II	40.6 ^a^	13.6 ^b^	13.2 ^b^	2.28	0.0002
H_Amide I	0.33 ^a^	0.13 ^b^	0.13 ^b^	0.019	0.0004
H_Amide II	0.21 ^a^	0.05 ^b^	0.03 ^b^	0.012	<0.0001
H_α-helix	0.32 ^a^	0.09 ^b^	0.12 ^b^	0.018	0.0002
H_β-sheet	0.28 ^a^	0.13 ^b^	0.13 ^b^	0.017	0.001
spectral ratio profiles					
A_Amide I/Amide II ratio	1.90 ^c^	4.29 ^b^	18.03 ^a^	0.394	<0.0001
H_Amide I/Amide II ratio	1.57 ^c^	2.59 ^b^	4.28 ^a^	0.175	0.0001
H_α-helix/β-sheet ratio	1.14 ^a^	0.71 ^c^	0.97 ^b^	0.007	<0.0001

^1^ SEM, standard error of the mean. Means with the different letters in the same row are significantly different (*p* < 0.05). ^2^ A_Amide I, amide-I area; A_Amide II, amide-II area; A_Amide I + II, amide-I + amide II area; H_Amide I, amide-I height; H_Amide II, amide-II height; H_α-helix, α-helix height; H_β-sheet, β-sheet height.

**Table 4 animals-12-00956-t004:** Correlation analysis between protein molecular structural spectral parameters ^1^ and nutritive values in different byproducts of seed-used pumpkin.

	A_Amide I		A_Amide II		H_Amide I		H_Amide II		H_α-Helix		H_β-Sheet		A_Amide I + II		A_Amide I/Amide II Ratio		H_Amide I/Amide II Ratio		H_α-Helix/β-Sheet Ratio	
	*r*	*p*	*r*	*p*	*r*	*p*	*r*	*p*	*r*	*p*	*r*	*p*	*r*	*p*	*r*	*p*	*r*	*p*	*r*	*p*
protein chemical profiles ^2^ (g/100g DM^−1^)																				
DM	−0.70	0.04	−0.56	0.12	−0.64	0.06	−0.17	0.66	−0.64	0.06	−0.58	0.10	0.06	0.87	−0.74	0.02	−0.63	0.07	−0.98	<0.0001
CP	0.95	0.0001	0.99	<0.0001	0.96	<0.0001	0.98247	<0.0001	0.95	<0.0001	0.95	<0.0001	0.97	<0.0001	−0.69	0.04	−0.83	0.006	0.73	0.03
SCP	0.95	0.0001	0.88	0.002	0.92	0.0004	0.89171	0.001	0.97	<0.0001	0.92	0.001	0.92	0.0004	−0.30	0.43	−0.51	0.16	0.94	0.0002
NPN	0.26	0.49	0.03	0.95	0.17	0.66	0.05868	0.88	0.29	0.46	0.19	0.62	0.16	0.69	0.66	0.05	0.47	0.20	0.62	0.08
ADICP	−0.61	0.08	−0.45	0.23	−0.54	0.14	−0.4679	0.20	−0.65	0.06	−0.53	0.14	-0.54	0.13	−0.30	0.43	−0.06	0.88	−0.95	0.0001
NDICP	0.01	0.98	0.22	0.58	0.10	0.80	0.1898	0.62	−0.03	0.94	0.10	0.80	0.10	0.79	−0.82	0.006	−0.65	0.06	−0.52	0.15
protein fractions partitioned by the CNCPS ^3^ (g/100g CP^−1^)																				
PA	−0.50	0.17	−0.69	0.04	−0.58	0.10	−0.67	0.05	−0.48	0.19	−0.57	0.11	−0.59	0.09	0.99	<0.0001	0.94	0.0002	−0.03	0.94
PB_1_	0.75	0.02	0.77	0.02	0.75	0.02	0.77	0.02	0.77	0.02	0.73	0.03	0.76	0.02	−0.39	0.30	−0.53	0.14	0.74	0.02
PB_2_	0.94	0.0001	0.96	<0.0001	0.95	<0.0001	0.96	<0.0001	0.96	<0.0001	0.94	0.0002	0.96	<0.0001	−0.57	0.11	−0.74	0.02	0.83	0.006
PB_3_	−0.20	0.61	−0.32	0.40	−0.26	0.51	−0.31	0.41	−0.15	0.70	−0.25	0.52	−0.25	0.51	0.70	0.04	0.64	0.06	0.22	0.56
PC	−0.65	0.06	−0.50	0.17	−0.58	0.10	−0.52	0.15	−0.69	0.04	−0.58	0.10	−0.59	0.10	−0.24	0.53	0.00	0.99	−0.96	<0.0001
in situ DM rumen degradation kinetics ^4^																				
*a* (g/kg)	0.49	0.18	0.31	0.42	0.41	0.27	0.33	0.38	0.53	0.1386	0.41	0.27	0.41	0.27	0.45	0.23	0.21	0.58	0.88	0.002
*b* (g/kg)	0.55	0.13	0.39	0.30	0.48	0.19	0.41	0.27	0.59	0.0925	0.47	0.20	0.48	0.19	0.35	0.35	0.11	0.78	0.92	0.0004
*a* + *b* (g/kg)	0.52	0.15	0.34	0.37	0.44	0.23	0.37	0.33	0.56	0.1161	0.44	0.24	0.44	0.23	0.41	0.28	0.17	0.66	0.90	0.0008
*c* (g/kg h^−1^)	0.91	0.0006	0.85	0.004	0.89	0.001	0.86	0.003	0.93	0.0003	0.88	0.002	0.89	0.001	−0.27	0.48	−0.48	0.19	0.93	0.0003
*ED* (g/kg)	0.54	0.13	0.37	0.32	0.47	0.20	0.39	0.29	0.59	0.0971	0.47	0.21	0.47	0.20	0.38	0.31	0.14	0.71	0.92	0.0005
in situ CP rumen degradation kinetics																				
*a* (g/kg)	0.40	0.28	0.21	0.59	0.32	0.40	0.23	0.55	0.45	0.23	0.32	0.40	0.32	0.40	0.54	0.14	0.31	0.42	0.83	0.006
*b* (g/kg)	0.62	0.07	0.48	0.19	0.56	0.12	0.50	0.17	0.67	0.05	0.55	0.12	0.56	0.11	0.26	0.51	0.01	0.97	0.96	<0.0001
*a* + *b* (g/kg)	0.53	0.14	0.36	0.35	0.45	0.22	0.38	0.32	0.57	0.11	0.45	0.22	0.45	0.22	0.39	0.29	0.16	0.69	0.91	0.0006
*c* (g/kg h^-1^)	0.89	0.001	0.81	0.008	0.86	0.003	0.82	0.006	0.92	0.0005	0.85	0.004	0.86	0.003	−0.18	0.64	−0.40	0.28	0.97	<0.0001
*ED* (g/kg)	0.57	0.11	0.40	0.28	0.50	0.17	0.42	0.26	0.61	0.08	0.49	0.18	0.50	0.17	0.35	0.36	0.11	0.78	0.93	0.0003

^1^ A_Amide I, amide-I area; A_Amide II, amide- II area; A_Amide I + II, amide-I + amide II area; H_Amide I, amide-I height; H_Amide II, amide-II height; H_α-helix, α-helix height; H_β-sheet, β-sheet height. ^2^ DM, dry matter; CP, crude protein; SCP, soluble crude protein; NPN, non-protein nitrogen; ADICP, acid detergent-insoluble crude protein; NDICP, neutral detergent-insoluble crude protein; ^3^ PA, non-protein nitrogen; PB_1_, rapidly degradable protein; PB_2_, intermediately degradable protein; PB_3_, slowly degradable protein; PC, bound protein. ^4^
*a*, rapidly degradable fraction in rumen degradation; *b*, slowly degradable fraction in rumen degradation; *a* + *b*, potentially degradable fraction in rumen degradation; *c*, degradation rate of the slowly degradable fraction; *ED*, effective degradability of the incubated samples.

**Table 5 animals-12-00956-t005:** Regression analysis to find the most important variables to predict nutrient supply using protein molecular spectral parameters in different byproducts of seed-used pumpkin.

Predicted Variable (*Y*) ^1^	Variable in the Model with *p* < 0.05 ^2^	Prediction Equations: *Y = a + b*_1_*x*_1_ *+ b*_2_*x*_2···....._	Model R^2^ Value	RSD ^3^	*p*-Value
Protein chemical profiles					
DM	H_amide I/amide II and H_α-helix/β-sheet left in the model	Y = 110.289 − 0.550 H_amide I/amide II − 18.934 H_α-helix/β-sheet	0.995	0.0701	<0.0001
CP (g/100g DM^−1^)	A_amide II left in the model	Y = 9.007 + 3.239 A_amide II	0.973	13.523	<0.0001
SCP (g/100g DM^−1^)	H_α-helix/β-sheet and H_α-helix left in the model	Y = −4.673 + 8.954 H_α-helix/β-sheet + 20.067 H_α-helix	0.993	0.135	<0.0001
ADICP (g/100g DM^−1^)	H_α-helix/β-sheet left in the model	Y = 21.250 − 19.319 H_α-helix/β-sheet	0.896	1.714	0.0001
NDICP (g/100g DM^−1^)	A_amide I/amide II left in the model	Y = 9.285 − 0.350 A_amide I/amide II	0.678	3.799	0.0064
protein subfractions using the CNCPS system					
PA (g/100g CP)	A_amide I/amide II left in the model	Y = 3.056 + 2.481 A_amide I/amide II	0.980	8.066	<0.0001
PB_1_ (g/100g CP)	H_α-helix left in the model	Y = 6.149 + 23.857 H_α-helix	0.588	5.594	0.0159
PB_2_ (g/100g CP)	H_α-helix/β-sheet and A_amide II left in the model	Y = −18.791 + 2.812 A_amide II-39.182 H_α-helix/β-sheet	0.972	21.238	<0.0001
PB_3_ (g/100g CP)	A_amide I/amide II left in the model	Y = 10.852 + 0.257 A_amide I/amide II	0.484	4.589	0.0374
PC (g/100g CP)	A_amide I/amide II and H_α-helix/β-sheet left in the model	Y = 138.971 − 120.225 H_α-helix/β-sheet-0.797 A_amide I/amide II	0.996	3.209	<0.0001
in situ DM rumen degradation kinetics					
*a* (g/kg)	A_amide I/amide II and H_α-helix/β-sheet left in the model	Y = −40.263 + 1.058 A_amide I/amide II + 83.074 H_α-helix/β-sheet	0.994	2.534	<0.0001
*b* (g/kg)	H_α-helix/β-sheet left in the model	Y = −27.820 + 61.952 H_α-helix/β-sheet	0.852	26.485	0.0004
*a* + *b* (g/kg)	A_amide I/amide II and H_α-helix/β-sheet left in the model	Y = −73.346 + 1.664 A_amide I/amide II + 145.417 H_α-helix/β-sheet	0.996	4.354	<0.0001
*c* (g/kg h^−1^)	H_amide I and H_α-helix/β-sheet left in the model	Y = 1.951 + 10.091 H_amide I + 7.775 H_α-helix/β-sheet	0.953	0.363	0.0001
*ED* (g/kg)	A_amide I/amide II and H_α-helix/β-sheet left in the model	Y = −63.890 + 1.350 A_amide I/amide II + 127.419 H_α-helix/β-sheet	0.996	3.289	<0.0001
in situ CP rumen degradation kinetics					
*a* (g/kg)	A_amide I/amide II and H_α-helix/β-sheet left in the model	Y = −20.660 + 0.987 A_amide I/amide II + 61.147 H_α-helix/β-sheet	0.989	2.704	<0.0001
*b* (g/kg)	H_α-helix/β-sheet left in the model	Y = −40.841 + 77.347 H_α-helix/β-sheet	0.918	21.107	<0.0001
*a* + *b* (g/kg)	A_amide I/amide II and H_α-helix/β-sheet left in the model	Y = −66.178 + 1.525 A_amide I/amide II + 138.842 H_α-helix/β-sheet	0.997	3.133	<0.0001
*c* (g/kg h^−1^)	H_amide I and H_α-helix/β-sheet left in the model	Y = −12.010 + 16.295 H_amide I + 21.082 H_α-helix/β-sheet	0.976	0.924	<0.0001
*ED* (g/kg)	A_amide I/amide II and H_α-helix/β-sheet left in the model	Y = −59.821 + 1.193 A_amide I/amide II + 123.848 H_α-helix/β-sheet	0.996	2.981	<0.0001

^1^ DM, dry matter; CP, crude protein; SCP, soluble crude protein; ADICP, acid detergent-insoluble crude protein; NDICP, neutral detergent-insoluble crude protein; PA, non-protein nitrogen; PB_1_, rapidly degradable protein; PB_2_, intermediately degradable protein; PB_3_, slowly degradable protein; PC, bound protein; *a*, rapidly degradable fraction in rumen degradation; *b*, slowly degradable fraction in rumen degradation; *a* + *b*, potentially degradable fraction in rumen degradation; *c*, degradation rate of the slowly degradable fraction; *ED*, effective degradability of the incubated samples. ^2^ A_Amide I, amide-I area; A_Amide II, amide- II area; A_Amide I + II, amide-I + amide II area; H_Amide I, amide-I height; H_Amide II, amide-II height; H_α-helix, α-helix height; H_β-sheet, β-sheet height. ^3^ RSD, residual standard deviation.

## Data Availability

The data presented in this study are available on request from the corresponding author.

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
