# Peer review of "Associations of Protein Molecular Structures with Their Nutrient Supply and Biodegradation Characteristics in Different Byproducts of Seed-Used Pumpkin"

_animals, 2022, doi:10.3390/ani12080956_

Round 1

Reviewer 1 Report

This manuscript represents an original approach to the topic. The results of special importance are:- there were significant differences in protein molecular structure and spectral parameters from pumpkin seed cake, pumpkin seed coat, and seed-used pumpkin flesh. The peak height ratios of the α-helix and β-sheet of the pumpkin seed coat were significantly lower than those of the pumpkin seed cake and seed-used pumpkin flesh; -correlations between the spectral parameters of seed-used pumpkin byproducts and their nutritional value, CNCPS protein subfractions and in situ rumen degradation characteristics; -multiple regression analyses used to select spectral parameters related to proteins to predict the protein subfraction, nutritional value, and rumen degradation characteristics of the seed-used pumpkin byproducts. This manuscript is also interesting from a methodological point of view, because it has been shown that FTIR could be used to evaluate the nutritional supply and biodegradation characteristics of different byproducts of seed-used pumpkin.

In previous research, the FTIR method has been used for the chemometric classification of pumpkin seed oils (for example: doi: 10.1016 / j.jbbm.2004.04.007. Or https://www.myfoodresearch.com/uploads/8/4/8/5 /84855864/_6__fr-2019-198_irnawati_2.pdf). When it comes to ruminants, classical experiments in nutrition have been done so far, where pumpkin seed was applied and its influence on production parameters in goats was examined (for example: DOI: https://doi.org/10.1017/S175173111700060X. There is no significant number of papers about proteins from these sources, and the available papers (for example: DOI: 10.1021 / jf402323u) talk about the extraction and characteristics of protein fractions, but not about the influence of their chemical composition on digestibility and utilization in cows. Due to the fact that FTIR was used as a method not only in the analysis of pumpkin seed but also in the process of biodegradation, which is designed to mimic the conditions of the rumen, this work has additional quality and contributes.

Combining different methods that allow to examine the biodegradability in rumen as a function of chemometric properties measured by FTIR methods represents a methodological contribution. In this paper, standard procedures for obtaining material were used. It may be necessary to describe the total number of samples more clearly.

Manuscript has a good fit even with a cursory reading. At start: “The purpose of this experiment was to explore the relationship of protein funtional groups (including amide I, amide II, α-helix, β-sheet) in byproducts of seed-used pumpkin (pumpkin seed cake, pumpkin seed coat, and seed-used pumpkin flesh) to their nutrient profiles and biodegradation characteristics.” At the end: “There were significant differences in the protein chemical profile, CNCPS subfraction, and rumen degradation kinetics of seed-used pumpkin byproducts. Among these byproducts, seed-used pumpkin flesh and pumpkin seed cake were shown to have high nutritional value and could provide a nitrogen source for ruminants. The significant differences in the chemical profile of CP and the degradability of byproducts of seed-used pumpkin were closely related to the changes in the molecular spectral parameters of the protein. The regression equation shows that the height ratio of α-helices to β-sheets was a strong predictor of the protein chemical profile and degradability. It was initially proven that FTIR could be used to evaluate the nutritional supply and biodegradation characteristics of different byproducts of seed-used pumpkin.” 

The quality of cited references is good, but we have only 20% of references younger than 5 years, but that did not affect my decision while reading the manuscript. However, if we look at the PubMed database when typing the keywords “pampkin seed protein”, we conclude that about 301 results have been published in the last 60 years, of which about 80 in the last 5 years. This could be the reason for the small number of recent references in the manuscript. Perhaps you can support the authors to, for the sake of future scientific communication, enrich the manuscript with a number of recent references.

In my opinion tables and figures are very informative equally from the aspect of fast reading of results as well as from the aspect of informative use of graphic solutions in the manuscript of scientific communication

Author Response

Response to Reviewer 1 Comments

Comments and Suggestions for Authors:

This manuscript represents an original approach to the topic. The results of special importance are: there were significant differences in protein molecular structure and spectral parameters from pumpkin seed cake, pumpkin seed coat, and seed-used pumpkin flesh. The peak height ratios of the α-helix and β-sheet of the pumpkin seed coat were significantly lower than those of the pumpkin seed cake and seed-used pumpkin flesh; -correlations between the spectral parameters of seed-used pumpkin byproducts and their nutritional value, CNCPS protein subfractions and in situ rumen degradation characteristics; -multiple regression analyses used to select spectral parameters related to proteins to predict the protein subfraction, nutritional value, and rumen degradation characteristics of the seed-used pumpkin byproducts. This manuscript is also interesting from a methodological point of view, because it has been shown that FTIR could be used to evaluate the nutritional supply and biodegradation characteristics of different byproducts of seed-used pumpkin.

In previous research, the FTIR method has been used for the chemometric classification of pumpkin seed oils (for example: doi: 10.1016 / j.jbbm.2004.04.007. Or https://www.myfoodresearch.com/uploads/8/4/8/5 /84855864/_6__fr-2019-198_irnawati_2.pdf). When it comes to ruminants, classical experiments in nutrition have been done so far, where pumpkin seed was applied and its influence on production parameters in goats was examined (for example: DOI: https://doi.org/10.1017/S175173111700060X. There is no significant number of papers about proteins from these sources, and the available papers (for example: DOI: 10.1021 / jf402323u) talk about the extraction and characteristics of protein fractions, but not about the influence of their chemical composition on digestibility and utilization in cows. Due to the fact that FTIR was used as a method not only in the analysis of pumpkin seed but also in the process of biodegradation, which is designed to mimic the conditions of the rumen, this work has additional quality and contributes.

Combining different methods that allow to examine the biodegradability in rumen as a function of chemometric properties measured by FTIR methods represents a methodological contribution. In this paper, standard procedures for obtaining material were used. It may be necessary to describe the total number of samples more clearly.

Manuscript has a good fit even with a cursory reading. At start: “The purpose of this experiment was to explore the relationship of protein funtional groups (including amide I, amide II, α-helix, β-sheet) in byproducts of seed-used pumpkin (pumpkin seed cake, pumpkin seed coat, and seed-used pumpkin flesh) to their nutrient profiles and biodegradation characteristics.” At the end: “There were significant differences in the protein chemical profile, CNCPS subfraction, and rumen degradation kinetics of seed-used pumpkin byproducts. Among these byproducts, seed-used pumpkin flesh and pumpkin seed cake were shown to have high nutritional value and could provide a nitrogen source for ruminants. The significant differences in the chemical profile of CP and the degradability of byproducts of seed-used pumpkin were closely related to the changes in the molecular spectral parameters of the protein. The regression equation shows that the height ratio of α-helices to β-sheets was a strong predictor of the protein chemical profile and degradability. It was initially proven that FTIR could be used to evaluate the nutritional supply and biodegradation characteristics of different byproducts of seed-used pumpkin.”

The quality of cited references is good, but we have only 20% of references younger than 5 years, but that did not affect my decision while reading the manuscript. However, if we look at the PubMed database when typing the keywords “pampkin seed protein”, we conclude that about 301 results have been published in the last 60 years, of which about 80 in the last 5 years. This could be the reason for the small number of recent references in the manuscript. Perhaps you can support the authors to, for the sake of future scientific communication, enrich the manuscript with a number of recent references.

In my opinion tables and figures are very informative equally from the aspect of fast reading of results as well as from the aspect of informative use of graphic solutions in the manuscript of scientific communication.

Response: We are truly grateful to yours and reviewers’ critical comments and thoughtful suggestions on our manuscript. Based on these comments and suggestions, we have made careful modifications on the original manuscript. All changes made to the text are in red color. All the comments/suggestions are responded carefully in detail.

In new manuscript (2.1. Sample collection, Lines 117-119), we had described the total number of samples more clearly. The new contents are: “The samples (n = 3) were collected from corresponding by-products, which were used as replicate.”

In addition, we supplemented the latest references to ensure the real-time nature of the article for the convenience of subsequent researchers. The number of references in the past five years has reached 18, accounting for 45%.

The new references:

  1. Keller, M.; Reidy, B.; Scheurer, A.; Eggerschwiler, L.; Morel, I.; Giller, K. Soybean meal can be replaced by faba beans, pumpkin seed cake, spirulina or be completely omitted in a forage-based diet for fattening bulls to achieve comparable performance, carcass and meat quality. Animals, 2021, 11, 1588.
  2. Fung, L.; Urriola, P.E.; Baker, L.; Shurson, G.C. Estimated energy and nutrient composition of different sources of food waste and their potential for use in sustainable swine feeding programs. Transl. Anim. Sci. 2019, 3, 359-368.
  3. Chen, Q.Q.; Lyu, Y.; Bi, J.F.; Wu, X.Y.; Jin, X.; Qiao, Y.N.; Hou, H.N.; Lyu, C.M. Quality assessment and variety classification of seed-used pumpkin by-products: Potential values to deep processing. Food. Sci. Nutr. 2019, 7, 4095-4104.
  4. Lyu, Y.; Bi, J.F.; Chen, Q.Q.; Wu, X.Y.; Qiao, Y.N.; Hou, H.N.; Zhang, X. Bioaccessibility of carotenoids and antioxidant capacity of seed-used pumpkin byproducts powders as affected by particle size and corn oil during in vitro digestion process. Food. Chem. 2021, 343, 128541.
  5. Dou, X.J.; Ma, Z.W.; Yan, D.; Gao, N.; Li, Z.Y.; Li, Y.; Feng, X.J.; Meng, L.X.; Shan, A.S. Sodium butyrate alleviates intestinal injury and microbial flora disturbance induced by lipopolysaccharides in rats. Food Funct. 2022, 13, 1360-1369.
  6. Shi, H.T.; Yu, P.Q. Comparison of grating-based near-infrared (NIR) and Fourier transform mid-infrared (ATR-FT/MIR) spectroscopy based on spectral preprocessing and wavelength selection for the determination of crude protein and moisture content in wheat. Food. Control. 2017, 82, 57-65.
  7. Liu, Q.; Zhang, W.; Zhang, B.; Du, C.G.; Wei, N.N.; Liang, D.; Sun, K.; Tu, K.; Peng, J.; Pan, L.Q. Determination of total protein and wet gluten in wheat flour by Fourier transform infrared photoacoustic spectroscopy with multivariate analysis. J. Food. Compost. Anal. 2021. 106, 104349.
  8. Guevara Oquendo, V.H.; Rodriguez Espinosa, M.E.; Yu, P.Q. Research progress on faba bean and faba forage in food and feed types, physiochemical, nutritional, and molecular structural characteristics with molecular spectroscopy. Crit. Rev. Food Sci. Nutr. 2021, 1-11.
  9. Zhang, G.N.; Li, Y.; Zhao, C.; Fang, X.P.; Zhang, Y.G. Effect of substituting wet corn gluten feed and corn stover for alfalfa hay in total mixed ration silage on lactation performance in dairy cows. Animal, 2021. 15, 100013.
  10. Hao, X.Y.; Yu, S.C.; Mu, C.T.; Wu, X.D.; Zhang, J.X. Replacing soybean meal with flax seed meal: effects on nutrient digestibility, rumen microbial protein synthesis and growth performance in sheep. Animal, 2020, 14, 1841-1848.
  11. Li, Y.; Zhang, G.N.; Fang, X.P.; Zhao, C.; Wu, H.Y.; Lan, Y.X.; Che, L.; Sun, Y.K.; Lv, J.Y.; Zhang, Y.G. Effects of replacing soybean meal with pumpkin seed cake and dried distillers grains with solubles on milk performance and antioxidant functions in dairy cows. Animal, 2021, 15, 100004.

Reviewer 2 Report

The work is well described, innovative and interesting for the expansion of databases of functional groups of proteins supported by correlations with conventional analyzes.
It would be advisable to format the entire bibliography following the guidelines of the journal and correct some formatting errors in the text.
Table 3 needs to be formatted.

Author Response

Response to Reviewer 2 Comments

Comments and Suggestions for Authors:

The work is well described, innovative and interesting for the expansion of databases of functional groups of proteins supported by correlations with conventional analyzes.

It would be advisable to format the entire bibliography following the guidelines of the journal and correct some formatting errors in the text.

Table 3 needs to be formatted.

Response: We are truly grateful to yours and reviewers’ critical comments and thoughtful suggestions on our manuscript. Based on these comments and suggestions, we have made careful modifications on the original manuscript. All changes made to the text are in red color. All the comments/suggestions are responded carefully in detail.

1 Introduction

Line 50 please, delete spaces between the references in the text.

The introduction is well written, short and concise and focused on the topics of the manuscript

Response: Spaces between the references in the introduction had been deleted.

2 Line 63 no space between [2, 3]

Response: Spaces had been deleted (Line 63).

3 Line 72-73 it is also rich in bioactive molecules like carotenoids and polyphenols (please, insert some references about the topic)

Response: new contents had been supplemented in new manuscript (Lines 72-74), and some references about the topic had been added. The new contents are: “Seed-used pumpkin flesh is rich in nutrients such as protein (114.0 g/kg DM), starch (51.6-102.4 g/kg DM), and soluble sugar (224.6-474.3 g/kg DM), crude fiber (130.5-185.7 g/kg DM) [1], and bioactive molecules (carotenoids, 0.20 ~ 19.57 mg/100g, wet basis) [5] which can provide sufficient energy and antioxidantsfor ruminants [6,7].”.

4 Line 81 no space

Response: Spaces had been deleted (Line 83).

5 Line 87 no space

Response: Spaces had been deleted (Line 90).

6 Line 88 no space

Response: Spaces had been deleted (Line 91).

7 Line 92 no space

Response: Spaces had been deleted (Line 95).

8 Line 151 please, create a new line (see template https://www.mdpi.com/journal/animals/instructions point 2 of Submission Checklist)

Response: According to the template you gave, we have modified the format of these formulas (Lines 157-161). I hope our modifications can meet your requirements, if you feel that there is anything that does not meet the requirements, please feel free to contact me. We will continue to revise.

9 Line 157 please, create a new line (see template https://www.mdpi.com/journal/animals/instructions point 2 of Submission Checklist)

Response: According to the template you gave, we have modified the format of these formulas (Lines 164-165). I hope our modifications can meet your requirements, if you feel that there is anything that does not meet the requirements, please feel free to contact me. We will continue to revise.

10 Line 182 please, create a new line (see template https://www.mdpi.com/journal/animals/instructions point 2 of Submission Checklist)

Response: According to the template you gave, we have modified the format of these formulas (Lines 189-194). I hope our modifications can meet your requirements, if you feel that there is anything that does not meet the requirements, please feel free to contact me. We will continue to revise.

11 Line 196 space before and afte the symbol <

Response: Space had been supplemented before and afte the symbol < (Line 208).

12 Line 219 space before and afte the symbol <

Response: Space had been supplemented before and afte the symbol < (Line 230).

13 Line 254 please, format the table.

Response: The format of the table has been modified (Lines 264-266).

14 Line 307 Discussion: please, delete space between the numbers of references in the text

Response: Spaces between the references in the Discussion had been deleted.

15 Line 445 The references should be in this form for Journal Articles:Author 1, A.B.; Author 2, C.D. Title of the article. Abbreviated Journal Name Year, Volume, page range. Please, format all the references, when it is necessary, as reported in the instructions

Response: Thank you very much for your comments, which have revised the formatting of all references and added some recent research literature.
